# Design Efficiency: A Critical Perspective on Testing Methods for Solar-Driven Photothermal Evaporation and Photocatalysis

**DOI:** 10.3390/nano15141121

**Published:** 2025-07-18

**Authors:** Hady Hamza, Maria Vittoria Diamanti, Vanni Lughi, Sergio Rossi, Daniela Meroni

**Affiliations:** 1Department of Chemistry, Università degli Studi di Milano, 20133 Milan, Italy; hadyahmed.hamza@unimi.it (H.H.); sergio.rossi@unimi.it (S.R.); 2Department of Chemistry, Materials and Chemical Engineering “Giulio Natta”, Politecnico di Milano, 20131 Milan, Italy; mariavittoria.diamanti@polimi.it; 3Department of Engineering and Architecture, Università degli Studi di Trieste, 34127 Trieste, Italy; vanni.lughi@dia.units.it; 4Consorzio Interuniversitario Nazionale per la Scienza e Tecnologia dei Materiali, 50121 Florence, Italy

**Keywords:** solar-driven steam, photothermal, photocatalysis, desalination, environmental remediation

## Abstract

Water scarcity is a growing global challenge, intensified by climate change, seawater intrusion, and pollution. While conventional desalination methods are energy-intensive, solar-driven interfacial evaporators offer a promising low-energy solution by leveraging solar energy for water evaporation, with the resulting steam condensed into purified water. Despite advancements, challenges persist, particularly in addressing volatile contaminants and biofouling, which can compromise long-term performance. The integration of photocatalysts into solar-driven interfacial evaporators has been proposed as a solution, enabling pollutant degradation and microbial inactivation while enhancing water transport and self-cleaning properties. This review critically assesses testing methodologies for solar-driven interfacial evaporators incorporating both photothermal and photocatalytic functions. While previous studies have examined materials and system design, the added complexity of photocatalysis necessitates new testing approaches. First, solar still setups are analyzed, particularly concentrating on the selection of materials and geometry for the transparent cover and water-collecting surfaces. Then, performance evaluation tests are discussed, with focus on the types of tested pollutants and analytical techniques. Finally, key challenges are presented, providing insights for future advancements in sustainable water purification.

## 1. Introduction

According to the Food and Agriculture Organization (FAO), approximately 3.2 billion people live in agricultural regions experiencing significant-to-severe water shortages or scarcity [1]. The worsening effects of climate change, seawater intrusion, and environmental pollution are further intensifying this crisis. Conventional water treatment methods, such as reverse osmosis, are highly energy-intensive [2]. Consequently, the development of energy-efficient technologies for converting polluted and saline water into potable water is an urgent necessity.

Solar-driven interfacial evaporators represent a promising low-energy and cost-effective solution. These systems function by harnessing solar energy to generate heat, accelerating the evaporation of water from wastewater, seawater, or brackish water. The resulting steam is then condensed in solar stills to produce purified water. In addition to their potential for providing clean water in emergencies and remote locations, solar stills have demonstrated economic viability compared to commercial water treatment methods for agricultural wastewater purification [3]. Other potential applications of solar-driven interfacial evaporators include energy generation and evaporative cooling [4]. Thus, these technologies contribute to a more sustainable and circular approach to natural resource management and pollution prevention and remediation.

Since 2016, solar distillation has gained renewed interest owing to evaporators based on interfacial heating [5]. These designs localize solar–thermal conversion near the evaporating interface, maintaining the bulk water cooler to reduce heat loss and enhance process efficiency. Typically, a photothermal absorber with high solar absorption and conversion efficiency is embedded in a porous, hydrophilic support that provides floatation and thermal insulation (Figure 1). Despite progress, the technology remains in its early stages. While various designs and materials have been explored, no definitive optimal solution has emerged [6,7]. Although much of the existing research focuses on issues such as salt rejection [8] and thermal losses [9], comparatively less attention has been given to challenges such as the inefficient removal of volatile and semivolatile contaminants, which can compromise water quality [10], and biofouling, which obstructs water transport channels in evaporators [11]. While several strategies have been proposed to address these concerns [12], the addition of photocatalytic materials in solar-driven interfacial evaporators has recently emerged as a promising and versatile solution [13].

The integration of a photocatalyst into the evaporator offers several advantages, including light-induced oxidation of organic pollutants and microbial inactivation. This results in higher-purity condensed water and reduced biofilm formation. Under irradiation, photocatalysts not only facilitate the degradation of organic contaminants but also promote the formation of a highly hydroxylated surface layer, thereby conferring superhydrophilic properties to the material [14,15]. These surface modifications enhance water transport while also inhibiting microbial adhesion and proliferation, extending the concept of self-cleaning to biological fouling prevention [16,17,18].

In general, many of the characteristics necessary for efficient photothermal catalysts align with those required for solar floating photocatalytic materials, facilitating the integration of these two solar technologies [13]. Consequently, integrating both processes into a single floating device holds significant promise, as it may provide an effective and energy-efficient alternative to current water remediation technologies. This dual-function approach enables the removal of both inorganic and organic contaminants while requiring significantly lower energy input and maintaining long-term performance.

However, despite these promising results, the lack of standardized testing methodologies often makes it difficult to compare and rationalize experimental findings. Therefore, this critical review provides a comprehensive analysis of testing techniques for solar-driven interfacial evaporators designed to integrate photothermal catalysts with photocatalysis. While previous reviews have extensively examined the geometries, materials, and operating principles of solar-driven interfacial evaporators [19,20,21,22,23], the incorporation of a photocatalytic component introduces additional complexity, requiring further methodological considerations. To date, to the best of the authors’ knowledge, these aspects have not yet been critically addressed. This review aims to bridge this gap by offering, for the first time, a critical perspective on the solar still design and materials, test pollutants, and protocols to be used to test the performance of solar-driven interfacial evaporators combining photocatalytic properties. First, the setup of solar stills used for hybrid solar-driven interfacial evaporators will be discussed, in particular concerning the transparent cover and water-collecting surfaces. The discussion of results from the literature will be complemented by original transmittance and wetting measurements. Then, performance testing methodologies will be discussed, also integrating the critical discussion of results from the literature with original photothermal measurements. Finally, key challenges and future research directions will be highlighted. While the discussion will focus on the combination of photothermal evaporation and photocatalysis, combinations of photothermal evaporation and photo-Fenton [24,25,26] or persulfate [27] processes have been recently reported, and some of the discussed issues apply also to approaches combining photothermal evaporation and other advanced oxidation techniques.

## 2. Materials for Photocatalysis and Photothermal Absorption

Table 1 provides an overview of the components used as photothermal catalysts and photocatalysts in hybrid solar-driven interfacial evaporators, together with the results of photothermal, photocatalytic, and combined performance tests. Each study reported in Table 1 used different actual irradiation conditions (see Section 3) and different test durations, so care should be taken in comparing different pollutant removal results. It should be noted that the aim of Table 1 is not to provide a direct performance ranking, but rather to highlight the diversity of materials, test pollutants, and experimental setups used across studies. This lack of standardization is a major barrier to cross-study comparisons and underscores the need for more consistent testing protocols.

Devices capable of simultaneously performing photothermal and photocatalytic functions can be engineered either by employing materials that convert light into both heat and reactive oxygen species or by immobilizing photocatalytic semiconductors together with photothermal agents [49,50]. When a single system is used, reduced complexity and potential cost-effectiveness are expected; while at least one such system has been investigated and shown promising results [51], this approach has intrinsic limitations, for the photogenerated carriers, the paths of charge separation or of heat generation are competitive for a given photon spectral range, and the system can hardly be optimized for both mechanisms. In most cases, the materials in which synergistic interactions between the photocatalytic and photothermal effects are claimed to occur are in fact hybrid systems at the nanoscale, where the individual components of the system are devoted to and optimized for either the charge transfer instrumental to photocatalysis or nonradiative recombination paths instrumental to thermal conversion [52]. When two separate systems are used, the higher complexity is compensated for by the possibility to maximize solar spectrum utilization, since high-energy ultraviolet (UV) photons are absorbed by the photocatalytic component and converted into high-redox-potential electron–hole pairs, while low-energy visible and near-infrared (NIR) photons are absorbed by the photothermal catalyst and transformed into heat [53]. This approach has been further developed by combining high-band-gap photocatalysts, like TiO_2_, with plasmonic nanoparticles, such as Ag. The latter not only act as photothermal catalysts by absorbing visible light through Localized Surface Plasmon Resonance (LSPR) but also enhance the photocatalytic performance of TiO_2_ by improving charge separation by acting as electron acceptors when titania is excited with UV light [54]. This approach has also been demonstrated for other systems [40].

## 3. Solar Still Geometry and Cover Material

### 3.1. Solar Still Geometry

Once the evaporator has been assembled, it needs to be thoroughly tested. Essential characterizations include its photothermal efficiency under a solar simulator (temperature mapping and evaporation rate) and the rejection of salts and volatile organic compounds. While tests of the former are carried out in a standard setup (Figure 2a), tests of the latter require a closed solar still, generally a single basin and of a passive type, enabling water collection. This design typically includes a basin filled with an aqueous solution containing the model salts or volatile organic compound, which is heated by sunlight, and a transparent cover or surface to condense and collect fresh water [55]. As the water evaporates, it condenses on the underside of the cover and flows into a collection channel [56]. There are numerous possible geometries and materials to be used for solar stills, as depicted in Figure 2b–f.

Flat-surface solar stills, the most common, use only one transparent cover or surface to condense and collect fresh water (Figure 2b). Single-surface solar stills are simple to construct, cost-effective, and ideal for small-scale water production in laboratory settings, when irradiation is provided by a lamp, or for outdoor applications, when irradiation is consistent [55]. The optimal cover angle for maximum yield is equal to the latitude angle of the location [57,58]. Double-surface solar stills enhance desalination efficiency by using two transparent surfaces (Figure 2c), allowing sunlight to enter from multiple angles and increasing the area for water condensation [59,60]. Another variant is the square pyramidal solar still (Figure 2d), where sloping sides allow sunlight to penetrate from multiple angles throughout the day, maximizing evaporation [61]. As alternatives to flat designs, hemispherical or dome-shaped stills (Figure 2e) and spherical solar stills (Figure 2f) maximize sunlight capture from multiple angles, enhancing thermal performance and evaporation throughout the day, as well as providing a large condensation surface. Hemispherical designs have been shown to outperform single-slope stills by up to 44% in terms of daily water production [62]. Adding reflectors has further boosted yields by up to 8%. However, dome designs are generally more complex and costly to construct than flat-surface models, and their curved surfaces can be harder to clean and maintain in an effective condition [63].

Independently of the geometry of the solar still, during pollutant rejection tests, a transparent cover is placed between the evaporator and the light source. The presence of this surface results in an effective irradiance loss and the filtering of certain wavelengths, due to the optical properties of the cover material. For this reason, it is essential to verify the actual spectrum and irradiance reaching the evaporator level, rather than simply reporting the type of lamp used. Moreover, during the test, water droplets tend to accumulate on the transparent cover surface, further reducing transmittance due to scattering phenomena.

The selection of appropriate materials and geometric configurations for the transparent cover and water-collecting surface thus plays a crucial role in optimizing the efficiency of solar-driven photothermal evaporation and photocatalysis. Several key factors influence this selection.

### 3.2. Optical Properties of the Cover Material

Numerous materials have been used as transparent covers for solar stills, including different types of glass, quartz, and polymers. Table 2 compares the optical properties of some commonly used commercial transparent materials.

In photocatalytic–photothermal evaporators, materials for the transparent cover should be chosen to align with the band-gap energy of the semiconductor to enhance absorption efficiency and optimize light-driven reactions. Figure 3a reports the light absorption features in the 200–1100 nm region for some of the commercial transparent materials, including both different types of glass and polymers. The material thicknesses were also matched as closely as possible based on commercially available sheets to ensure consistent comparison across samples. The AM1.5D standard solar spectrum is reported as a reference [90].

Some of the most used photocatalysts have band gaps ≥ 3.0 eV, such as TiO_2_ (absorption edge: 390 nm for anatase, 410 nm for rutile), ZnO (absorption edge: 390 nm), and BiOCl (375 nm). Hence, the choice of the transparent cover is particularly crucial when large-band-gap semiconductors are to be used, due to their need for UV irradiation to activate the photocatalytic process. To this end, Figure 3b compares the spectra of the tested materials in the 200–400 nm range, reporting also the AM1.5D reference solar spectrum.

Quartz has a higher transmittance across the whole UV region, but its higher costs and more difficult workability make it a less palatable option than glass. Indeed, inexpensive and readily available materials are preferred to ensure economic feasibility and scalability for real-world applications. Among glass types, borosilicate (Pyrex), low-iron (or optical), and soda–lime glass were compared. They differ in the cutoff wavelength in the UV region, with borosilicate showing the largest window of UV transmittance among the three. However, the UV component of the solar spectrum at ground level is in the wavelength range 300–400 nm, so mostly in the UV-A range. As their cutoff wavelengths are below 320 nm, all types of glass represent suitable choices even when large-band-gap photocatalysts are to be used. As reported in Table 1, such photocatalysts are indeed frequently employed in the literature. It should be noted, however, that increasing the thickness generally causes a lower transmittance [91], which can shift the cutoff value to longer wavelengths. In the visible and NIR spectrum, borosilicate glass and optical glass achieve a higher light transmittance compared to standard soda–lime glass, due to their low iron content.

Optically transparent polymers could represent a cheap, mechanically resistant, and easier-to-shape alternative to glass and quartz. Moreover, their compatibility with 3D printing techniques such as stereolithography enables the fabrication of complex geometries with high precision. In addition, 3D printing allows for rapid prototyping and cost-effective, material-efficient production of transparent covers specifically designed for solar still applications. This approach also supports decentralized, on-demand manufacturing, which is particularly valuable in remote or resource-limited settings.

In general, polymer layers offer a limited UV transmittance, due to the strong UV absorption below 300 nm of several organic moieties (carbonyl, substituted aromatic rings) present in either the polymer chain or additives in commercial sheets. In polymers such as PET, crystallinity introduces haze, reducing transmittance in the visible region due to light scattering in crystalline domains. Organic moieties also induce intense absorption peaks in the near- and mid-IR region (e.g., C=O stretching, aromatic C–H stretching, ester group vibrations, O-H overtones). Furthermore, it should be considered that commercial polymeric sheets often contain UV-absorbing additives to promote outdoor durability, which shift their absorption edge toward the visible range. The same issue is often observed for 3D-printable polymers by stereolithography due to UV light-absorbing activators (Figure 4).

Overall, the solar still design should minimize light transmittance losses, not only by the careful choice of the cover material, but also by optimizing the shape and thickness of the material. Cover material plays a key role also in water vapor condensation, as will be discussed in the next section.

### 3.3. Wetting Properties of the Cover Material

During solar still operation, water vapor tends to condensate at the cover surface. Water droplet formation on the cover surface can negatively impact light transmittance, leading to reduced energy absorption and overall efficiency. This issue is particularly critical in photocatalysis due to its inherently low quantum yield.

The wetting properties of the cover material determine the condensation beginning time, number of water molecules condensed over time, and accumulation rate [92]. A lower contact angle is beneficial to promote faster condensation. Figure 5 compares the wetting features of some common cover materials. All tests were carried out under the same light irradiation, at constant temperature and environmental humidity, using the transparent covers described in Figure 3. Optical glass (Figure 5a) displays the lowest contact angle. Other types of glass and quartz generally show similar, highly hydrophilic behavior. Conversely, polymeric materials (Figure 5b–d) display higher contact angles, indicative of poorly hydrophilic (PMMA and PET) and even hydrophobic properties (PP).

Moreover, by changing the shape of the droplet, wetting properties affect the light transmittance across the wet cover material. In the case of glass, lower contact angles lead to the spreading of the droplets (Figure 5a). On the other hand, water condensates in small, clearly defined droplets on polymeric materials (Figure 5b–d). This leads to marked differences in terms of the light transmittance of the wet surfaces (Figure 6).

Condensate formation invariably leads to a loss of transmittance across the whole UV-vis-NIR range due to scattering. In addition, water absorption bands further contribute to reduced transmittance in the IR region. However, the extent of transmittance loss varies considerably with the material, and it is linked to its wetting features. When small droplets are formed, as in the case of poorly hydrophilic and hydrophobic surfaces, scattering is predominant and leads to a marked loss in transmittance. Conversely, highly hydrophilic materials, such as glass, tend to form a water film upon condensation at their surface, limiting the scattering impact. In the case of a superhydrophilic surface, a homogeneous film is formed, preserving visual clarity [93,94].

Indeed, the modification of the wetting properties of the cover material is a key strategy for enhancing water condensation in solar-driven interfacial evaporators [20]. Superhydrophilic coatings have been shown to promote condensation while mitigating light obstruction. Conversely, superhydrophobic coatings can improve vapor collection by reducing droplet adhesion, although they can impede condensation. Biomimetic approaches, such as patterned superhydrophilic/superhydrophobic surfaces inspired by the Namib beetle [95], offer promising avenues for optimizing water condensation performance in cover materials. It should also be considered that light transmittance and reflection from a solar still cover play a critical role in thermal management and overall system performance. For further insights into the complex challenges of thermal management in solar-driven interfacial photothermal evaporators, readers are encouraged to consult recent comprehensive reviews on the subject [20].

Considering performance, scalability, and costs, presently, borosilicate and optical glass are the optimal choices for solar still distillation applications. They are cheaper and more widely available than quartz, making them suitable for cost-effective large-scale deployment. They are resistant to UV irradiation, thermal stress, and exposure to volatile organic compound vapors. Their high light transmittance in the 300–1400 nm range ensures effective solar energy absorption by the solar evaporator to drive water photocatalytic and photothermal effects in the still. Compared to polymers, their highly hydrophilic nature enables the fast condensation of the water vapor and spreading of the condensed droplets into films, preserving light transmittance and facilitating water collection.

## 4. Target Pollutants and Testing Protocols

Table 2 also reports the performance of photocatalytic–photothermal devices reported in the literature in the removal of model water pollutants. Photothermal tests (column 7) refer to experiments conducted using only the photothermal component, without the photocatalyst, where the concentration of the pollutant in distilled water is measured (data reported as the distillate concentration ratio, i.e., the ratio between the concentration in the distilled phase and the initial concentration in the water to be treated). Photocatalytic tests (column 8) involve monitoring the change in pollutant concentration in the liquid phase under irradiation, using either only the photocatalyst or the full hybrid device; pollutant removal data are reported as the ratio between the final pollutant concentration in the treated water and the initial concentration. Photothermal + photocatalytic tests (column 9) are performed using the complete photothermal/photocatalytic device; results are reported as the distillate concentration ratio. The starting pollutant concentration reported in column 6 corresponds to the highest pollutant concentration that allows for comparison across different types of tests performed in the same paper.

Comparability among results is limited by the different types of setups used during the tests (e.g., light cover, distance between evaporator and light source) as well as their duration and test conditions (type of pollutant, concentration, water matrix, etc.).

Here, we will instead focus only on the appropriate testing procedures to assess the removal of organic contaminants and rejection of volatile organic compounds, which are the most specific factors to the case of photothermal–photocatalytic solar-driven interfacial evaporators.

As clearly shown in Table 2, there is no agreed-upon standard testing procedure to compare photocatalytic–photothermal evaporators, also in terms of model pollutants, and most studies report only VOC rejection tests.

The first issue is the choice of target pollutant(s) to be tested. In the literature, a variety of target molecules have been used to test the performance of photothermal–photocatalytic solar-driven interfacial evaporators, as reported in Table 1. The most commonly tested molecules include dyes (such as methylene blue, rhodamine B, Congo red, malachite green, acid fuchsin, eriochrome black T, Bengal rose B sodium salt, methyl red, methyl orange) and aromatics (including phenol, toluene, aniline, 4-nitrophenol, nitrobenzene). Emerging pollutants, such as pharmaceuticals and personal care products like metronidazole and ciprofloxacin, are more rarely investigated.

As mentioned in the Introduction, the rationale to investigate the combination of photocatalysis with photothermal evaporation lies in the fact that volatile/semivolatile compounds tend to accumulate in distilled water. In this respect, the degradation of organic compounds in the water basin can lead to their lower accumulation in the condensed phase. However, not all molecules are equal in this respect.

Figure 7 compares the distillate concentration ratio (*R*_D_), calculated using Equation (1), of different organic contaminants upon photothermal evaporation in a solar still.*R*_D_ = *C*_D_/*C*_I_(1)
where *C*_D_ is the pollutant concentration in the condensed water and *C*_I_ is the initial concentration in the wastewater to be treated [28]. The *R*_D_ parameter is used to evaluate the contaminant concentration in the distillate: a lower *R*_D_ value indicates higher rejection of the pollutant by the photothermal effect.

As Figure 7 clearly shows, methyl orange is effectively rejected from the distilled water by the photothermal effect (*R*_D_ = 0). This observation can be traced back to its low volatility and applies to most organic dyes. Conversely, both bisphenol A and phenol can be found in the distilled water. However, while bisphenol A displays a partial rejection (*R*_D_ << 1), phenol preferentially accumulates in the condensate (*R*_D_ ca. 1). This difference may be related to the different volatilities of the different substances. Chen and coworkers [10] demonstrated that the distillation of volatile/semivolatile compounds during interfacial solar distillation under ambient atmospheric pressure is mainly governed by the compounds’ Henry’s law constant, *K*_H_. They showed that compounds less volatile than phenol were effectively rejected by photothermal distillation. However, above that threshold, they observed a negative correlation between *R*_D_ and *K*_H_, since more-volatile compounds cannot easily achieve their saturation concentration in the air and condensate less in the distilled water, leaving a significant fraction of uncondensed molecules in the air.

In the present case, the Henry’s law constant at 25 °C of bisphenol (4.0 × 10^−11^ atm·m^3^·mol^−1^) is lower than that of phenol (6.29 × 10^−7^ atm·m^3^·mol^−1^), explaining the lower observed *R*_D_ value. The volatility of azo dyes is even lower; hence, their concentration in the distillate is negligible since they do not migrate in the gas phase.

Despite their popularity as test pollutants in studies in the literature, dyes are thus not suitable benchmarks for testing the effectiveness of photothermal and photocatalytic–photothermal evaporators in rejecting volatile compounds for water remediation. They should be used as target pollutants only to evaluate the photocatalytic performance of the photocatalyst embedded within the photocatalytic–photothermal evaporator. Even though the use of dyes as test molecules for photocatalysis is debated [96], they are still widely used in the photocatalysis literature due to their ease of monitoring. In photocatalytic–photothermal evaporators, using dyes as model compounds in photocatalytic tests offers the advantage of being able to disentangle the photothermal and photocatalytic contributions, as any decrease in the dye content of the liquid phase cannot be attributed to photothermal evaporation. This is important because it can aid in optimizing the photocatalytic performance of the device (e.g., tuning the composition of the photocatalyst, ensuring effective irradiation of the photocatalyst, etc.) and in evaluating its efficiency in the challenging conditions of photothermal tests (especially in terms of electrolyte content). Given the complex structure and high surface areas of the porous materials used in photothermal devices, photocatalytic characterization should also include dark adsorption tests and recyclability tests.

On the other hand, to assess the rejection of volatile organic compounds, compounds with high vapor tension should be used, such as phenol [43], 4-nitrophenol, and nitrobenzene, measuring their concentration in the distilled water. In particular, phenol should be considered a benchmark, owing to it having the highest potential to contaminate distilled water thanks to its high volatilization and condensation. This second test is key to ensure the purity of evaporated water in a solar evaporator and should be performed in parallel with tests of the photocatalytic efficiency of the device, as previously reported by some authors [37,43,44]. It should be noted that many papers focus only on a single compound (or a single class of compounds) or report just the concentration of pollutants in the distilled water, often failing to acknowledge the fundamental difference in the probing power of the two tests.

We thus propose that the two single-pollutant tests should be routinely reported in all studies proposing new evaporator designs. In addition to testing with individual model compounds, future studies should also address the complexity of multi-pollutant systems, where synergistic or competitive transport mechanisms may arise, potentially affecting both contaminant rejection and photocatalytic efficiency.

Finally, we would like to stress that the potential for contamination is an important factor, particularly when volatile organic compounds are targeted in purification processes. Phenols in particular can react with polymer components. Hence, not only the transparent cover material but also the material of the solar still and water-collecting bottle should be chosen taking into consideration chemical inertness. The reactivity of the inert, floating support, or wicking components used in tests should also be evaluated.

## 5. Conclusions and Prospects

In this review, we clarified the most suitable test setups for evaluating photocatalytic–photothermal evaporators and provided guidelines for selecting the transparent cover material based on the specific type of photocatalyst employed. We advocate that the choice of solar still geometry and materials should take into account scalability by

(1)Prioritizing low-cost, widely available materials (e.g., optical glass over quartz);(2)Designing for manufacturability: While flat stills are cheaper and more versatile to fabricate, their lower efficiency highlights the need for cost-effective solutions to produce dome stills. While 3D printing offers design flexibility, currently available transparent printable materials present significant limitations in optical performance.

We also proposed a dual-testing strategy that should always be conducted in parallel to comprehensively assess a device’s performance. The first test studies photocatalytic performance using dyes as target pollutants, monitoring their concentration in the feedwater. The second instead evaluates the rejection of volatile organic compounds (VOCs) by tracking the concentration of phenol, as a model VOC, in the distillate.

Several open questions remain and warrant further investigation. The actual mechanisms governing these systems are complex, involving simultaneous and potentially synergistic contributions from photocatalysis and photothermal evaporation. Fundamental studies aimed at disentangling the different pathways at play are needed to guide the design of more-efficient systems.

Inorganic electrolytes, such as chlorides and bicarbonates, are known to adversely affect the photocatalytic performance of various photocatalysts, including TiO_2_ [97]. Since solar-driven interfacial evaporators are intended to operate with saline feeds, such as seawater, the interplay between electrolytes in the water matrix and the photocatalyst in photocatalytic–photothermal evaporators should be systematically investigated. Various strategies can be envisaged to address this challenge, such as introducing ionic shielding layers, employing photocatalysts with reduced sensitivity to electrolytes, or localizing the photocatalyst in areas of the evaporator exposed to lower electrolyte concentrations.

Finally, future testing protocols should incorporate real-world contaminants, including regulated priority pollutants, to ensure that performance assessments are aligned with environmental standards and application-specific requirements. This approach will enhance the relevance and impact of laboratory findings in practical water treatment scenarios.

## Figures and Tables

**Figure 1 nanomaterials-15-01121-f001:**
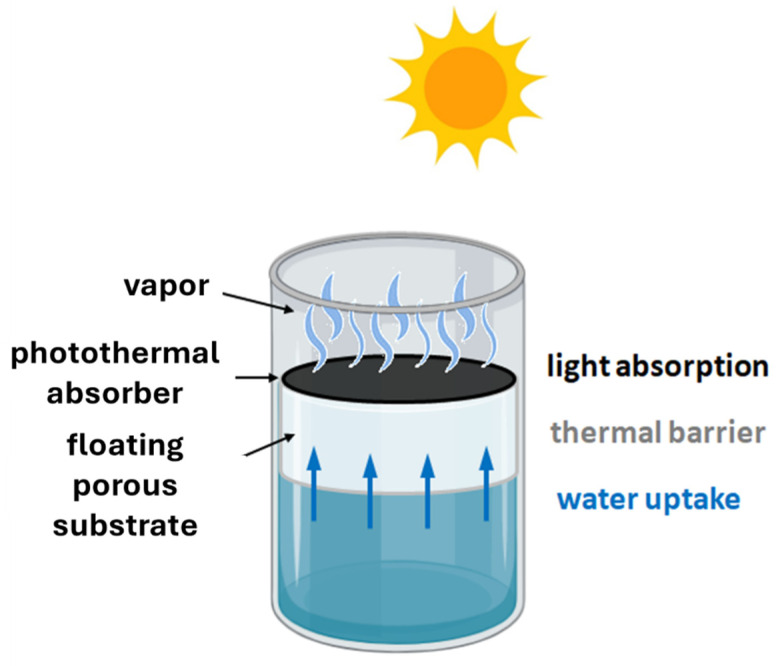
General structure of solar-driven interfacial evaporators.

**Figure 2 nanomaterials-15-01121-f002:**
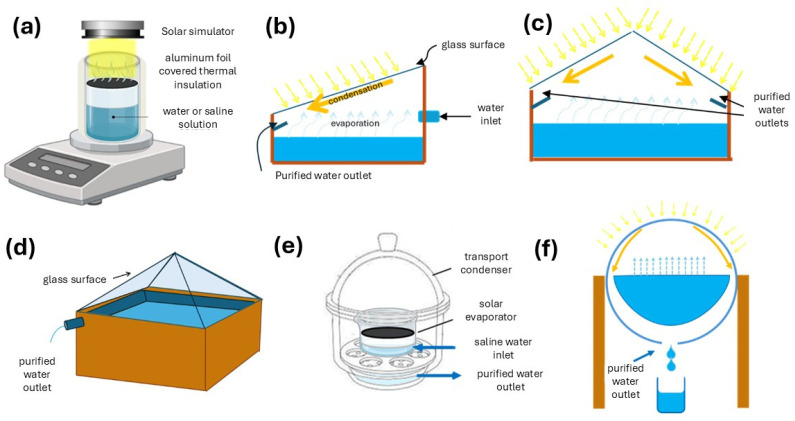
(**a**) Setup for photothermal efficiency determination under solar simulator; (**b**–**f**) different designs of solar stills for tests of rejection of salts and volatile organic compounds: (**b**) single-surface solar still, (**c**) double-surface solar still, (**d**) square pyramidal surface solar still, (**e**) dome-shaped still, and (**f**) spherical solar still.

**Figure 3 nanomaterials-15-01121-f003:**
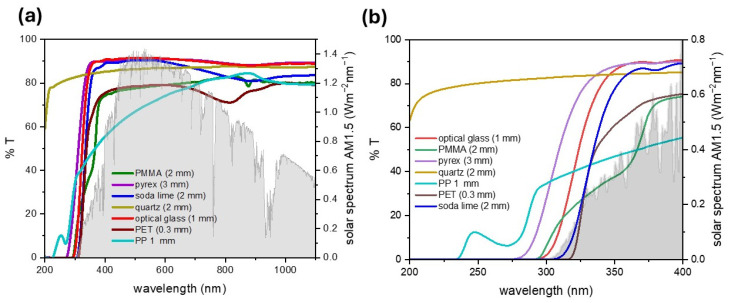
(**a**) Transmittance in the UV-vis-NIR region of a series of transparent materials commonly used in solar stills. The thickness of the flat sheets used for determination is given in parentheses. The AM1.5D solar spectrum is reported as a reference [90]. (**b**) Zoomed-in view of the UV region (200–400 nm).

**Figure 4 nanomaterials-15-01121-f004:**
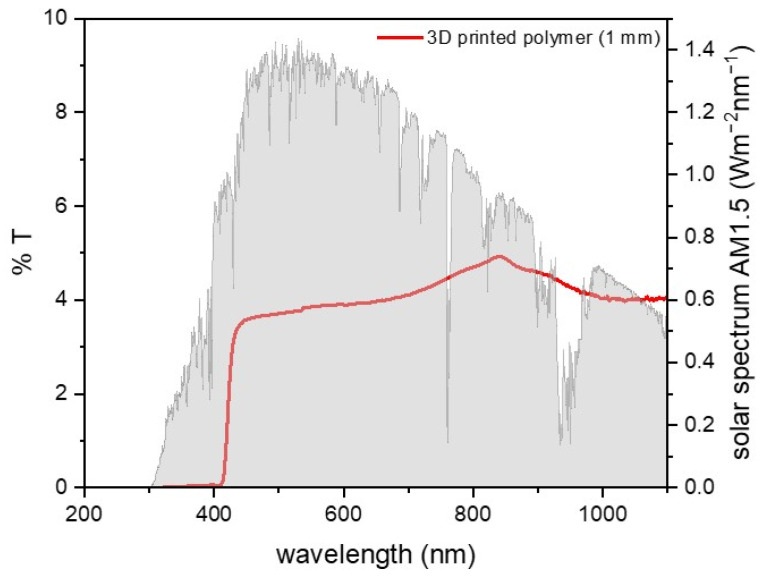
UV-vis spectrum of the clear V4 resin sample (Formlabs), composed of trimethylolpropane triacrylate, bisphenol A ethoxylate diacrylate, and isobornyl acrylate, with (2,4,6-trimethylbenzoyl)phosphine oxide as the radical photoinitiator. The sample was 3D-printed using a Formlabs Form3 printer with Legacy settings and a layer thickness of 0.050 mm. The printed part, with a total thickness of 1 mm, was gradually polished using sandpaper up to 3000 grit. The AM1.5D solar spectrum is reported as a reference [90].

**Figure 5 nanomaterials-15-01121-f005:**
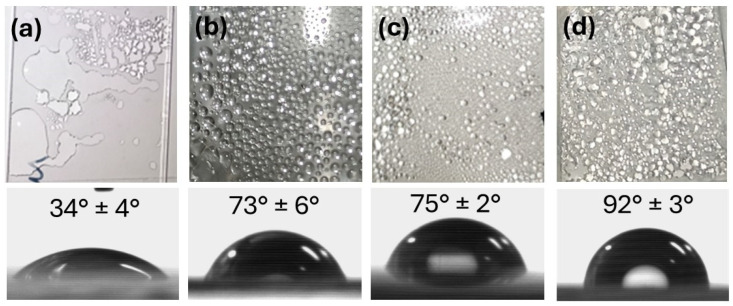
Water contact angle and images of water condensate on a transparent sheet (all with the same tilt angle of 30°) of optical glass (**a**), PMMA (**b**), PET (**c**), and PP (**d**).

**Figure 6 nanomaterials-15-01121-f006:**
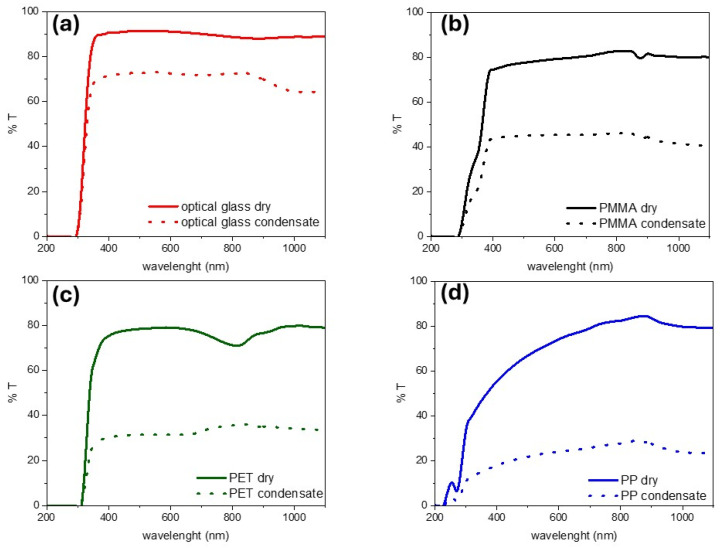
Effect of formation of water condensate on the transmittance spectra of a transparent sheet of glass (**a**), PMMA (**b**), PET (**c**), and PP (**d**).

**Figure 7 nanomaterials-15-01121-f007:**
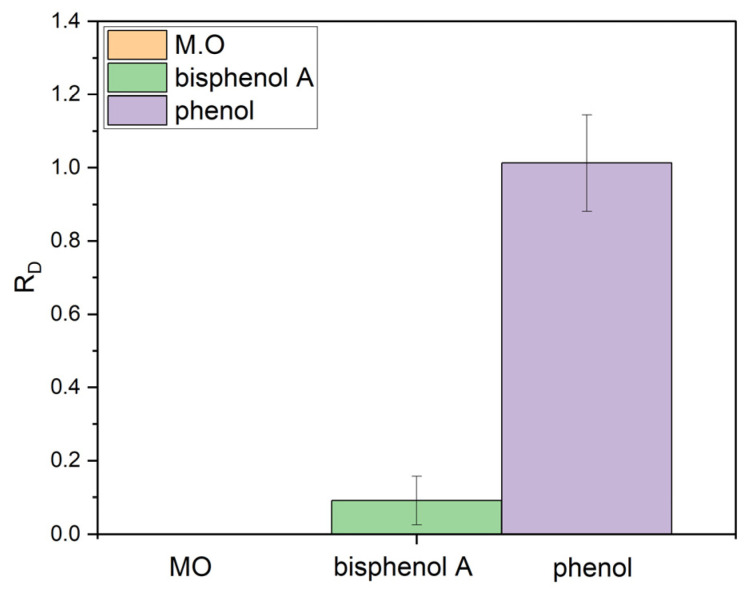
Distillate concentration ratio (*R*_D_) of different organic contaminants (20 ppm in water) upon photothermal evaporation in a solar still.

**Table 1 nanomaterials-15-01121-t001:** List of photothermal evaporators integrating photocatalytic materials, with their performance in the removal of a series of organic pollutants. A dash is used to denote missing or unavailable data.

Organic Pollutants	Photothermal Component	Photocatalyst	Support	Solar Still Design	Tested Concentration (mg L^−1^)	VOC Removal			Ref.
						Photothermal (No Evaporator)—*R_D_*	Only Photocatalysis—*C_final_*/*C*_0_ in Treated Liquid	Photothermal + Photocatalytic—*R_D_*	
phenol	calcined *Saccharum spontaneous*	ZnFe_2_O_4_	polyethylene foam	single solar still	10	90	-	5	[28]
4-nitrophenol						100	-	4	
nitrobenzene						90	-	10	
thionine	iron phthalocyanine organic polymer	CoFe_2_O_4_ and polydopamine	melamine foam	beaker/single solar still	-	-	0	-	[29]
methylene blue						-	0	-	
rhodamine B						-	0	-	
phenol					1	-	-	1	
*N*,*N*-dimethylformamide						-	-	10	
aniline						-	-	7	
methylene blue	polyimide/MXene	polypyrrole/BiVO_4_	thermal insulation material (cotton)	hemispherical still	20	-	21	-	[30]
rhodamine B						-	29	-	
Congo red						-	23	-	
malachite green						-	19	-	
methyl orange	MOF-derived C/TiO_2_		expanded polyethylene foam	beaker	10	-	-	-	[31]
rhodamine B						-	12	-	
tetracycline	black g-C_3_N_4_/chitosan		Hydrogel and polyethylene foam	double-surface solar still	100	-	17	-	[32]
methyl orange	Au nanoparticles@TiO_2_@Pt		-	Pyrex glass, open reactor	10	-	2	-	[33]
methyl red						-	2	-	
methylene blue						-	0	-	
rhodamine B	Au@TiO_2_		microporous membrane	beaker	20	-	46	-	[34]
rhodamine B	Au@TiO_2_		anodized aluminium oxide membrane	beaker (type of cover not shown)	20	-	40	0	[35]
methyl orange	TiO_2_–polydopamine@polypyrrole		Cotton	beaker	10	-	4	-	[36]
rhodamine B	MoO_3−x_/ BiOCl/CNTs		cellulose acetate membrane	beaker	5	-	-	0	[37]
toluene				not shown	10	52	-	0	
methyl orange	SnSe@SnO_2_		-	not shown	10	-	0	0	[38]
rhodamine B	Au@hierarchical ZnO		mixed cellulose ester membrane	beaker	15	-	30	0	[39]
phenol	TiO_2_/Ti_3_C_2_/C_3_N_4_/polyvinyl alcohol		Hydrogel	single solar still	10	117 *	0	15 *	[40]
phenol	BiOBr_0.85_I_0.15_		Melamine sponge	dome-shaped still	10	78	10	0	[41]
phenol	carbonized carboxymethyl chitosan/alginate hydrogel crosslinked by Cu^2+^		polyethylene foam	beaker in quartz close container	10	-	-	3	[18]
p-chlorophenol						-	-	ca. 5	
p-methylphenol						-	-	ca. 5	
rhodamine B	TiO_2_-CuO		Cu foam	dome-shaped still	10	-	-	13	[42]
phenol						164	-	20	
phenol	graphene/polypyrrole aerogels		polystyrene foam	dome-shaped still	20	148	17	7	[43]
ciprofloxacin					10	-	7	1	
methyl orange						-	5	1	
rhodamine B						-	4	0	
mixture					20	-	-	10	
phenol	β-MnO_2_/porous hydrogel		polyurethane sponge	N.A.	20	50	28	1	[44]
methylene blue						1	73	1	
methyl orange						1	71	1	
rhodamine B	Ti_3_C_2_ MXene/CdS		polystyrene (PS) foam	double-jacketed beaker	10	-	0	-	[45]
phenol				Petri dish and inverted beaker		116	-	15	
metronidazole						31.5	-	0	
phenol	TiO_2_		porous clustered carbon array foams	reactor with horizontal cover	5	200	-	20	[46]
quinol						120	-	40	
aniline						210	-	40	
phenol	Cu/W_18_O_49_@graphene		polydimethylsiloxane (PDMS) sponge	dome-shaped still with quartz cover	10	120	-	1	[47]
phenol	m-TiO_2−x_ nanofibrous membrane		polystyrene foam	dome-shaped still with quartz cover	10	105	-	4.5	[48]

* Calculated from total organic carbon (TOC) values.

**Table 2 nanomaterials-15-01121-t002:** Optical properties of several commercial materials usable in solar stills.

	UV	Visible	IR	References
**Quartz**	up to 90% transmittance across most of the UV spectrum, including UV-C	up to 92–95% transmittance	90–95% transmittance till 2500 nm and up to 4000 nm in some types	[1,64,65,66]
**Soda–Lime Glass (Standard Window Glass)**	40–75% UV-A transmittance; 1–10% transmittance of UV-B and UV-C	ca. 85–90% transmittance	good transmittance in 750–2500 nm, it blocks >2500 nm	[64,67,68,69]
**Low-Iron Glass (Optical or Extra-Clear Glass)**	higher UV-A transmittance than soda–lime glass but still significantly limits UV-B and UV-C	ca. 91–93% transmittance	similar to soda–lime glass but with slightly more transmission in 750–1400 nm range	[70,71,72]
**Borosilicate Glass (Pyrex)**	75–90% transmittance in UV-A; higher transmittance in the UV-B range than optical glass; blocks UV-C	ca. 85–90% transmittance	high transmittance in 750–1500 nm range, it blocks >2000 nm	[73,74,75,76]
**Laminated Glass**	<1% transmittance across the UV spectrum	70–88% transmittance	20–50%, depending on glass thickness and type of interlayer used	[77]
**Poly(methyl methacrylate) (PMMA)**	up to 75% transmittance in the 300–400 nm range	up to 90% transmittance	good transmittance in 700–2500 nm but with strong absorption bands in mid-IR due to C–H and C=O vibrational modes	[78,79,80,81,82]
**Polyethylene Terephthalate (PET)**	moderate transmittance >360 nm	ca. 85% transmittance (amorphous form) ca. 80% transmittance (semi-crystalline form)	moderate transmittance in NIR, poor transmittance in mid- and far-IR	[83,84,85,86]
**Polypropylene (PP)**	10–30% transmittance in the UVA range	semi-transparent	moderate transmittance in near-IR but with strong overtone absorption peaks	[87,88,89]

## Data Availability

Research data are available at the following https://doi.org/10.5281/zenodo.15878429 (accessed on 19 November 2024).

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
