# Peer review of "Design Efficiency: A Critical Perspective on Testing Methods for Solar-Driven Photothermal Evaporation and Photocatalysis"

_nanomaterials, 2025, doi:10.3390/nano15141121_

Round 1
Reviewer 1 Report
Comments and Suggestions for Authors
This study reports a critical perspective on testing methods for solar-driven photothermal evaporation and photocatalysis. However, the manuscript needs to be further improved and polished before possible publication.
1.
- The introduction requires revision. There should be an additional discussion of the literature review. Highlight the limitations of the existing literature (what is the gap in the literature, especially water collection efficiency), i.e, Materials Science and Engineering: R: Reports 165, 101018; Small 21 (7), 2407280; Desalination 603, 118655; Advanced Functional Materials, 2509989; and Advanced Science, 2415101 can be cited. Emphasize the uniqueness and novelty of the present study.
- The review offers a broad and technically rich overview, but its structure could benefit from improved organization. A clearer thematic grouping would enhance readability.
- It is suggested that the author conduct a review on thermal management, comparing the achievements of different researchers and analyzing their respective characteristics. For example, some examples could be provided in terms of solar energy utilization and improvement of thermal losses.
- It has also come to my attention that the role of condensers of different materials is missing, comparing their performance, analyzing their advantages, and disadvantages.
- The review mentions that many systems are difficult to scale. What are the primary barriers to scale-up e.g. material cost, fabrication complexity, or system integration and how do proposed solutions address these?
Author Response
Please see the responses in the attachment.

Reviewer 2 Report
Comments and Suggestions for Authors
The review titled “Design Efficiency: A critical perspective on testing methods for solar driven photothermal evaporation and photocatalysis” clarifies the testing methodologies for interfacial solar evaporators incorporating both photothermal and photocatalytic functions. It also proposed a dual testing strategy to comprehensively assess the device’s performance. This review provides unique insights into future advancements in sustainable water purification. However, the following issues still need to be modified before they can be considered for publication.
- The phrases of “solar-driven interfacial evaporators”, “solar evaporators”, “solar-driven water evaporators” and “interfacial solar evaporator” should be named consistently throughout this review.
- In the Introduction, the excessive seven paragraphs can be appropriately reduced, for example, the third paragraph that introduces the advantages of “solar evaporators” and the fourth paragraph that introduces the shortcomings of “solar evaporators” at this stage can be summarized into one paragraph.
- In section 3, the author can summarize the current application status by citing published articles on the practical application of “solar evaporators” in the treatment of salts and volatile organic compounds. For example, ACS Nano, 2025, 19(12): 11625–11647, Mater. 2024, 36(5): 2303976, Nano Energy 2018, 46, 415–422, Adv. Mater. 2022, 34, 2110548, and Nano Lett. 2024, 24, 11615–11623.
- In Figure 2, the size of the text in the charts should be ensured to be uniform and legible.
- The style of Figure 6 is not harmonized with Figure 4.
- Some grammatical problems, such as “the device’s performance” instead of “the device performance” to be corrected.
Author Response

(The authors gave the same response as above.)

Reviewer 3 Report
Comments and Suggestions for Authors
This manuscript effectively provides a systematic review of test methods for solar-driven interfacial evaporation and photocatalytic synergistic water purification systems, with a focus on assessing the current lack of standardization and suggesting innovative ideas for material selection, pollutant testing strategies, and more. Although this manuscript is complete in structure and clear in organization, several important issues are still needed to be solved before it can be considered for publication. Therefore, I suggest that this article can be published, but the paper needs major revision before acceptance for publication. My detailed comments are as follows:
1. The first abbreviated form should have its full name, for example, FAO.
2. The final conclusion section suggests adding some discussions on prospects, and the title of this section should also be changed to "Conclusion and Prospects".
3. The literature review in the manuscript was not comprehensive enough. Although some references were cited to support the arguments, the coverage of key literature in the field of interfacial evaporation was insufficient, such as https://doi.org/10.1016/j.greenca.2024.10.003; Energy Mater, 2024;4:400021; Advanced Functional Materials‚ 2025‚ 345, 2504823; Research, 2024, DOI: 10.34133/research.0347. For example, when discussing photothermal and photocatalytic materials, some important research findings were not mentioned, which may have led to a lack of completeness and objectivity in the manuscript's conclusions.
4. The data analysis of the manuscript was not in depth enough, and although a large amount of experimental data were listed, the analysis and comparison of them were relatively blank. For example, for the experimental results of the organic pollutant removal performance in Table 1, there was no systematic comparison and discussion of these data, and it failed to study in depth the reasons for the differences in pollutant removal efficiency under different materials and experimental conditions.
5. The experiment on water droplet accumulation, water contact angle and light transmittance in Figure 5 of the manuscript was the highlight of the manuscript, but some parameters were missing, and it should be added whether the light intensity, ambient humidity and material thickness were completely unified in this experiment.
6. The unsuitability of dyes (e.g., methyl orange) for testing volatile pollutant interception (due to low volatility) was noted in Section 4 of the manuscript, but coexisting interferences (e.g., electrolytes affecting photocatalysis) of alternatives (e.g., phenolics) in real water bodies were not discussed, and experimental evidence of multi-pollutant synergistic transport needed to be added.
7. The manuscript was insufficiently expansive on the challenges facing the current key issues, such as the inhibition of photocatalytic activity by chlorides and bicarbonates mentioned in the conclusion of Section 5, but no solutions (e.g., material surface modification, ionic shielding layers) were provided, which could be further supplemented by research programs in recent years.
Author Response

(The authors gave the same response as above.)

Round 2
Reviewer 3 Report
Comments and Suggestions for Authors
Although the manuscript has undergone some revisions, almost all the figures in the current manuscript are produced from other articles. As a good review paper, appropriate original pictures are needed. For instance, regarding the integration of solar evaporation and photocatalysis, it is also recommended that there be relevant original figure in the outlook section.
Author Response
We appreciate the Reviewer’s feedback. However, we would like to clarify that all the figures included in the manuscript are original and specifically created for this work. Among the seven figures presented, five are based on original experimental or simulation data produced by the authors. To enhance clarity and transparency, we have now deposited the relevant datasets in a public repository and added a reference to it in the revised manuscript. We have also revised the Introduction to more clearly state the extent to which the figures are original and based on our own data.
Regarding the suggestion to include an original figure on the integration of solar evaporation and photocatalysis in the Outlook section, we note that this discussion is currently included within the Conclusion and Prospects. We believe that adding an additional figure at this stage would not substantially improve the clarity or depth of the discussion. Nevertheless, we have ensured that the conceptual integration is clearly addressed in the text. We trust that these clarifications help to resolve the concern.